# Assessing community engagement in Nigeria polio eradication initiative: application of the Consolidated Framework for Implementation Research

Oluwaseun Oladapo Akinyemi [1], Adedamola Adebayo,[2] Christopher Bassey,[2] Chioma Nwaiwu,[2] Anna Kalbarczyk [3], Akinola A Fatiregun,[4] Olakunle O. Alonge,[3] Eme Owoaje[2]

¹Health Policy and Management, University of Ibadan College of Medicine, Ibadan, Oyo State, Nigeria
²Department of Community Medicine, University of Ibadan College of Medicine, Ibadan, Oyo, Nigeria
³International Health, Johns Hopkins University Bloomberg School of Public Health, Baltimore, Maryland, USA
⁴Ondo State Field Office, World Health Organization Country Office for Nigeria, Ondo, Nigeria

**Correspondence to**
Dr Oluwaseun Oladapo Akinyemi;
seunakinyemi@hotmail.com

## ABSTRACT

**Objective** This study employed the Consolidated Framework for Implementation Research (CFIR) to assess factors that enhanced or impeded the implementation of community engagement strategies using the Nigerian polio programme as a point of reference.

**Design** This study was a part of a larger descriptive cross-sectional survey. The CFIR was used to design the instrument which was administered through face-to-face and phone interviews as well as a web-based data collection platform, Qualtrics.

**Setting** The study took place in at least one State from each of the six geopolitical zones in Nigeria (Nasarawa, Borno, Kano, Sokoto, Anambra, Bayelsa, Lagos, Ondo and Oyo States as well as the Federal Capital Territory).

**Participants** The respondents included programme managers, policy-makers, researchers and frontline field implementers affiliated with the Global Polio Eradication Initiative (PEI) core partner organisations, the three tiers of the government health parastatals (local, state and federal levels) and academic/research institutions.

**Results** Data for this study were obtained from 364 respondents who reported participation in community engagement activities in Nigeria's PEI. Majority (68.4%) had less than 10 years' experience in PEI, 57.4% were involved at the local government level and 46.9% were team supervisors. Almost half (45.0%) of the participants identified the process of conducting the PEI program and social environment (56.0%) as the most important internal and external contributor to implementing community engagement activities in the community, respectively. The economic environment (35.7%) was the most frequently reported challenge among the external challenges to implementing community engagement activities.

**Conclusion** Community engagement strategies were largely affected by the factors relating to the process of conducting the polio programme, the economic environment and the social context. Therefore, community engagement implementers should focus on these key areas and channel resources to reduce obstacles to achieve community engagement goals.

## INTRODUCTION

Community engagement is a fundamental principle in the planning, organising and delivery of primary healthcare services to the community.[1 2] This principle, set during the Alma-Ata declaration of 1978, remains relevant in the Sustainable Development Goals era when engaging the community is the priority for the achievement of universal health coverage by 2030.[3] The benefits of engaging the community in the planning and implementation phases of health interventions have been documented across various maternal and child health programmes, including the polio eradication programme.[4–6] In the primary healthcare system, this strategy supports relationship building with community members,[7] reduces health inequalities,[8] provides a channel for the dissemination of information on health products and services and aids the health system to adapt the health service to the local context and address misinformation about health interventions.[9 10]

## Strengths and limitations of this study

► This study presents findings of one of the most important strategies of the polio eradication programme and presented findings which are useful for public health programme implementers in achieving community engagement objectives.

► The participants in this study are a representative of all categories of stakeholders from the national to the local levels that participated in the polio programme thereby bringing in their nuanced insights.

► Social desirability bias was a possibility because participants were involved in the programme and would probably not be inclined to present the activities in a negative light.

► This study used a cross-sectional design; therefore, it was impossible to determine causal relationships between variables. Furthermore, there was a tendency to recall bias as the Polio Eradication Initiative spanned a few decades.

The implementation of community engagement strategies in the polio programme has enhanced coverage, acceptability and ownership of health programmes in high-risk areas such as hard-to-reach and conflict-affected communities, which would have otherwise been difficult to achieve.[5] Community engagement has been beneficial and has encouraged the adoption of health services by individuals who have negative beliefs and attitudes by engaging them to discuss the basis of their beliefs and encouraging them to proffer solutions to issues they identified.[11] One successful example of community engagement in high-risk areas includes the 2014 engagement of 11 000 female community-based mobilisers in Nigeria to advocate for vaccine uptake in northern Nigeria.[12] This resulted in over 322 000 newborns referred for routine immunisation and over 32 000 malnourished children referred for appropriate care.[12]

Alternatively, when community engagement was not used and these high-risk communities were neglected during planning and implementation of public health interventions, misconceptions about the programme emerged leading to noncompliance to the public health directives or outright rejection of health interventions.[13] The polio programme in Nigeria has had first-hand experience with vaccine hesitancy and rejection of the polio vaccines. This was a result of neglecting political and religious leaders' opinions and not addressing their misconceptions before programme implementation, especially in Northern Nigeria.[13 14] This challenge was eventually resolved through dialogue with religious and political leaders and by engaging the community in validating the vaccine.[6] The Global Polio Eradication Initiative (GPEI) then adopted this strategy in addressing other challenges that arose in these communities and in other countries yet to be certified polio-free.[15–17] The community engagement strategy has taken various formats over the years ranging from engaging religious leaders and traditional leaders to engaging opinion leaders and other members of the community. Other community engagement strategies include creating media awareness as well as providing incentives to caregivers and other participating community members. These innovations have yielded substantial benefits to the immunisation system and the primary healthcare system.[4 18]

The wild poliovirus was last detected in Nigeria in 2016, and community engagement has been identified as a principal strategy for achieving this success.[12] However, as most community engagement activities in polio have been reactionary, there is a gap in our understanding of how to systematically translate best practices in community engagement activities to other programmes and contexts. Whereas other Primary Healthcare programmes have leveraged on the gains of the Polio Eradication Initiative (PEI) in Nigeria and are adopting the strategies employed by the polio programme, including community engagement, the adoption of these strategies has yielded mixed results due to variable implementation—and lack of a systematic model to adapt these strategies to other contexts. Therefore, there is a need to extensively discuss potential factors that could hinder or facilitate the implementation of community engagement strategies using the polio programme in Nigeria as a case study.

The Consolidated Framework for Implementation Research (CFIR),[19] which provides relevant domains and constructs for analysing public health strategies, can be applied to systematically identify relevant aspects for the implementation of community engagement strategies, without which, such strategies would be unsuccessfully implemented in any context. This study used the CFIR to identify factors that facilitated or hindered the implementation of community engagement activities within the Nigeria's PEI. The study was conducted as part of a multi-country implementation science project, Synthesis and Translation of Research and Innovations from Polio Eradication, to capture lesson learnt from the GPEI.[20] It is hoped that the knowledge of these factors will enhance the adaptation of community engagement activities in other public health programmes, and the effective delivery of these programmes in Nigeria and similar settings.

## METHODS
### Study setting
Nigeria is the most populous country in sub-Saharan Africa with an estimated population of 200 million persons of which over two-thirds are children and young persons.[21] Nigeria is divided into six geopolitical zones namely South-South, South-East, South-West, North-East, North-Central and North-West. Overall, Northwest, Kano-Sokoto hub, have the highest number of cases of poliomyelitis followed by the Northeast, Borno-Yobe hub.[22]

In recent times, Nigeria has experienced security challenges initially from the militants in the south and insurgents and armed bandits in the northern part of the country. This has put health workers engaged in vaccination efforts in jeopardy.[23 24] Consequently, the country missed the polio-free certification in 2016 when a case of polio was reported in the North East region (Borno), one of the areas affected by conflict and insecurity.[18] However, it has been 3 years since the last case of the wild poliovirus was identified in Nigeria[25] and the country was eventually declared polio-free by the WHO in June 2020.[26]

### Study design and participants
Three hundred and sixty-four participants, who were part of a larger descriptive survey comprising of 1020 participants, were purposively selected to participate in this study. Participants in the larger study were recruited through emails, telephone calls and in-person contact. These means of recruitment were used to ensure the estimated sample of participants was achieved. The research team approached potential participants through the state ministry of health, the executive secretary to the minister of health to some states (Ondo and Oyo) provided a list of health workers who participated in polio programme in his state. This list

includes state immunisation officers, surveillance officers, technical facilitators to the immunisation. For the remaining states, the research team made contact with focal persons at health institutions to recruit health workers for the survey.

This study used the CFIR as an evaluation framework to identify critical challenges and contributors in the implementation of community engagement activities within the PEI in Nigeria. The CFIR is an evaluation tool for assessing implementation activities, challenges and contributors. The CFIR was selected because it is a multitheoretical framework that has been used to synthesise research evidence from various disciplines into a consolidated framework with multiple constructs of what works across different contexts and why they work.[19 27] The framework has 39 constructs organised into five major domains found to influence the successful implementation of innovative programmes. The domains assessed include Characteristics of the innovation; Outer setting including the economic, political and social context; Inner setting including organisational culture and climate; Characteristics of individuals involved in the implementation; and Process of conducting the intervention.[27]

The participants were selected from nine states across the six geopolitical zones in Nigeria and the Federal Capital Territory. The nine states, namely Nassarawa, Borno, Kano, Sokoto, Anambra, Bayelsa, Lagos, Ondo and Oyo States and the Federal Capital Territory Abuja, were purposively selected based on the existing rapport between the research team and contact persons/facilitators in these states. The respondents for this study were eligible to participate in the study based on their engagement in PEI activities for at least 1 year since its inception in 1988. The respondents included programme managers, policymakers, researchers and front-line field implementers affiliated with the GPEI core partner organisations, the three tiers of the government health parastatals and academic/research institutions. The respondents were identified across the three tiers of government in Nigeria—federal, state (subnational) and local government (district). At the federal level, key stakeholders in the PEI were recruited from the National Primary Healthcare Development Agency, Federal Ministry of Health and GPEI core partners group including WHO, UNICEF and Bill & Melinda Gates Foundation. At the state level, PEI managers, supervisors, team leads, and facilitators were recruited. At the local government level, front-line implementers in the polio programme were recruited.

### Survey instrument
Development of the survey instrument was informed by activities of the PEI programme, tools used by the GPEI partners to generate lessons learnt from the polio programme and attempts to fill gaps left from previous efforts as well as external efforts. Key GPEI programme activities domains (and related implementation

challenges) were identified in a literature review of GPEI literature.

The survey instrument was a structured closed-ended questionnaire with sections on demographics and participants' experiences with implementing community engagement activities. The survey instrument was pretested among 10 Resident Doctors at the University College Hospital, Ibadan by the research team to ensure that questions were properly asked and adequately answered by the research assistants and respondents, respectively. These respondents provided face content and validity for the questionnaires.

The participants' experiences with implementing community engagement activities were organised into two categories:
1. Facilitators—These were the factors that contributed to the success in the implementation of community engagement activities in the PEI, Nigeria.
2. Barriers—These were the factors that impeded the implementation of community engagement activities in the PEI, Nigeria.

For the two categories, the CFIR framework was used to inform the responses. The CFIR framework contains domains and constructs which had the potential to serve as facilitators or barriers in implementing community engagement activities (table 1).

### Data collection procedures
The survey instrument was administered online using Qualtrics, a web-based data collection and analysis platform. The self-administered survey was in English language. The Qualtrics link was sent via email to potential respondents with instruction to respond within 2 weeks. Subsequently, the research team sent an email as a reminder to the potential respondents 2 weeks after the initial contact. The survey was conducted between September 2018 and January 2019.

Potential participants who had difficulty in participating online either as a result of a busy schedule or technical difficulties had the option of a face-to-face interview or telephone interviews. These interviews were conducted by research assistants who had at least a Bachelor's degree. All the survey data were uploaded by a research assistant on the Qualtrics platform.

### Data analysis
Data were first exported from the Qualtrics platform into SPSS statistical software package, V.20. Data were then cleaned and frequencies were used to summarise the demographic characteristics. Domain analysis of the CFIR was conducted to the highest occurring factor that contributes/and impedes successful community engagement. This was further broken down into internal and external domains contributors. While for barriers to community engagement. CFIR domain analysis was conducted to identify the most significant domain and further analysis was conducted on construct level to gain more insight into these barriers.

**Table 1** CFIR domains and associated constructs

| Construct | | Short description |
|---|---|---|
| **Internal setting** | | |
| **I.** | **Intervention/PEI programme characteristics** | |
| A | Intervention source | Perception of key stakeholders about whether the intervention is externally or internally developed |
| B | Evidence strength and quality | Stakeholders' perceptions of the quality and validity of evidence supporting the belief that the intervention will have desired outcomes. |
| C | Relative advantage | Stakeholders' perception of the advantage of implementing the intervention versus an alternative solution. |
| D | Adaptability | The degree to which an intervention can be adapted, tailored, refined or reinvented to meet local needs. |
| E | Trialability | The ability to test the intervention on a small scale in the organisation, and to be able to reverse course (undo implementation) if warranted. |
| F | Complexity | Perceived difficulty of implementation, reflected by duration, scope, radicalness, disruptiveness, centrality and intricacy and no of steps required to implement. |
| G | Design quality and packaging | Perceived excellence in how the intervention was bundled, presented and assembled. |
| H | Cost | Costs of the intervention and costs associated with implementing the intervention including investment, supply and opportunity costs. |
| **II.** | **Inner/organisational settings** | |
| A | Structural Characteristics | The social architecture, age, maturity and size of an organisation. |
| B | Networks and communications | The nature and quality of webs of social networks and the nature and quality of formal and informal communications within an organisation. |
| C | Culture | Norms, values, and basic assumptions of a given organisation. |
| D | Implementation climate | The absorptive capacity for change, shared receptivity of involved individuals to an intervention, and the extent to which use of that intervention will be rewarded, supported, and expected within their organisation. |
| E | Readiness for implementation | Tangible and immediate indicators of organisational commitment to its decision to implement an intervention—leadership engagement and resources available. |
| F | Others | Other personal attributes |
| **III.** | **Characteristics of individuals within your organisation involved in PEI activities** | |
| A | Knowledge and beliefs about the intervention | Individuals' attitudes toward and value placed on the intervention as well as familiarity with facts, truths, and principles related to the intervention. |
| B | Self-efficacy | Individual belief in their capabilities to execute courses of action to achieve implementation goals. |
| C | Individual stage of change | Characterisation of the phase an individual is in, as he or she progresses toward skilled, enthusiastic, and sustained use of the intervention. |
| D | Individual identification with organisation | A broad construct related to how individuals perceive the organisation, and their relationship and degree of commitment with that organisation |
| E | Other personal Attributes | A broad construct to include other personal traits such as tolerance of ambiguity, intellectual ability, motivation, values, competence, capacity and learning style. |
| **IV. Process of conducting the activities** | | |
| A | Planning | The degree to which a scheme or method of behaviour and tasks for implementing an intervention are developed in advance, and the quality of those schemes or methods. |
| B | Engaging | Attracting and involving appropriate individuals in the implementation and use of the intervention through a combined strategy of social marketing, education, role modelling, training and other similar activities. |

| | Construct | Short description |
|---|---|---|
| C | Executing | Carrying out or accomplishing the implementation according to plan. |
| D | Reflecting and evaluating | Quantitative and qualitative feedback about the progress and quality of implementation accompanied with regular personal and team debriefing about progress and experience. |
| **V.** | **External/outer setting** | |
| A | Political environment | Lawmaker support, political climate accepting of polio eradication activities, and political structure to conducive to coordinated action |
| B | Economic environment | Sufficient revenue sources/base to fund activities and/or maintain system developments |
| C | Social environment | Social norms around immunisation, accepting communities in which polio eradication activities were implemented |
| D | Technological environment | Infrastructure or technological advances outside of the organisation |
| E | Other environment | Environment where activity was implemented was prohibitive and did not contribute to the success of polio eradication, including the global climate and ineffective cross-organisational collaboration |

Adapted from a study[16] on Fostering implementation of health services research findings into practice: A Consolidated Framework for advancing Implementations Science.
CFIR, Consolidated Framework for Implementation Research; PEI, Polio Eradication Initiative.

## Patient and public involvement

Patients and/or the public were not involved in the design, or conduct, or reporting, or dissemination plans of this research.

Written informed consent was obtained from the study participant who participated in the face-to-face interviews while verbal consent was obtained from participants interviewed via the telephone after the objectives and rationale of the study had been explained. Participants who provided their responses via the Qualtrics platform signed their consent electronically after a detailed description of the study and its rationale was provided. Participants were informed that their responses were anonymous unless they indicated that they would like to be contacted for follow-up and dissemination of survey results. They were assured that their information will remain confidential and will not be shared beyond the research team without their consent.

## RESULTS
### Sociodemographic characteristics

About 249 (68.4%) participants had been engaged in the Nigerian PEI for less than 10 years. The respondents had been engaged at the various levels of implementation of PEI activities; more than half (57.4%) of the respondents worked at the subdistrict/local government level. The most common roles included team supervisors (46.9%) and front-line health workers (30.8%). Additional sociodemographic characteristics of survey participants are presented in table 2.

### Factors that contributed to the successful implementation of community engagement activities in the PEI, Nigeria

A total of 45.0% of the participants reported that the process of conducting the PEI programme (eg, planning of stakeholder engagement, engagement of appropriate stakeholder, executing the activities as planned and monitoring engagement outcomes against the stated objectives) was the most important internal contributor to implementing community engagement activities in the community. This was followed by other internal factors 24.0% including characteristics of individuals within the organisation involved in the PEI activities, awareness of benefits of community engagement, health workers enthusiasm and sustained support throughout the stages of the engagement and commitment to the organisation and PEI programme characteristics 21.0% including the positive perception of quality and effectiveness of community engagement activities, the adaptability of the strategy to local context, ease of implementation of the strategy, and minimal challenges encountered during implementation (see figure 1).

The most frequently mentioned external contributor to the community engagement activities was the social environment, 56.0%, in which the community engagement activity was implemented (particularly sociocultural beliefs around immunisation). Also, the political factors (stakeholders' and political support) within the communities 26.0% was a commonly mentioned external contributor. The economic factors, 16.0%, were also mentioned as contributors to community engagement activities (figure 2).

### Factors that challenge the implementation of community engagement activities in the PEI, Nigeria

The external environment (50%) such as stakeholders' lack of interest, poor funding mechanisms and norms that hinder immunisation campaigns, was the most implicated factor in the challenges to community engagement activities. This is followed by the process of conducting

**Table 2** Sociodemographics characteristics of respondents

| Respondents characteristics | n (%) |
|---|---|
| **Work experience (years)** | |
| <10 | 249 (68.4) |
| 10–19 | 96 (26.4) |
| ≥20 | 19 (5.2) |
| **Level of involvement*** | |
| District | 209 (57.4) |
| State | 168 (46.2) |
| Subdistrict/local government | 120 (32.9) |
| National | 64 (17.6) |
| Global | 21 (5.8) |
| **Affiliated organisation*** | |
| Government agencies | 348 (96.6) |
| Other GPEI Partners | 234 (64.3) |
| UNICEF | 147 (40.4) |
| Implementing partners | 163 (44.8) |
| Global NGO | 20 (5.5) |
| Academic research | 5 (1.4) |
| **Roles with the PEI programme in Nigeria*** | |
| Team supervisor | 171 (46.9) |
| Front line health worker | 112 (30.8) |
| Monitoring and oversight board | 78 (21.4) |
| Strategy committee/management group | 57 (15.6) |
| Programme officer | 37 (10.2) |
| Surveillance officer | 35 (9.6) |
| EPI manager | 32 (8.8) |
| Programme manager | 25 (6.9) |
| Policy-maker | 12 (3.3) |
| Researcher | 7 (1.9) |
| Country project lead | 6 (1.6) |
| Others† | 101 (27.7) |

*Multiple response questions.
†Others include: consultant, cluster coordinator, data analysis officer, medical officer of health.
GPEI, Global Polio Eradication Initiative; NGO, Non-Governmental Organisation.

the community engagement activities (23.1%) and the characteristics of people involved in the polio eradication activities (12.4%) such as lack of awareness of benefits of the vaccine, poor knowledge of principles of vaccination (see figure 3).

The respondents indicated that within the external environment domain, the economic environment (35.7%) such as funding mechanism and the social environment (32.9%) such as sociocultural beliefs about vaccination, were the most frequent external challenges to community engagement. Other challenges highlighted were the

political environment (26.9%) such as political will and set priorities and the technological environment (9.3%) which includes access to technology. The global climate and ineffective cross-organisational collaborations were mentioned by 29.1% of the respondents as challenging factors in the category of other environments.

As shown in table 3, of all the processes involved in conducting PEI programme activities, the process of conducting community engagement (46.4%) such as types of activities and stakeholders involved and how the activities were organised was the most important challenge to implementing community engagement strategies. This was closely followed by the execution phase (44.0%), the planning phase (35.7%) and the reflection phase (27.4%).

In the domain exploring the characteristics of individuals involved in the PEI activities within the organisation, the individual's knowledge about the importance of community engagement and belief that community engagement may not be effective (62.2%) was the most reported challenge in terms of effectively engaging community members during the implementation of PEI activities. Other factors identified in this category were the individual's perception about and degree of commitment to the PEI (28.0%), the individual's stage of change (26.8%), self-efficacy (13.4%), and other personal attributes (4.9%) (table 3).

According to respondents, the level of readiness for implementation within the PEI organisation construct (46.7%) was the most frequently recorded challenge in the domain of conducting Intervention/PEI programme characteristics. This was closely followed by the level of networking and communication within the organisation (35.0%). The culture, implementation climate and structural characteristics within the PEI organisation were the least reported challenges to community engagement (28.3%, 26.7% and 18.3% respectively) (table 3).

## DISCUSSION

This paper identified the process of conducting community engagement activities (such as planning of stakeholder engagement, engagement of appropriate stakeholder, executing the activities as planned and monitoring engagement outcomes against intended objectives) as a major contributor to the success of community engagement strategy under the PEI programme in Nigeria. Characteristics of individuals (for instance awareness of benefits of the community engagement, health workers enthusiastic and sustained support throughout the stages of the community engagement and commitment to the organisation) implementing the activities were identified as another contributor. While the external environment, especially the economic and social environment (eg, sociocultural beliefs that hinder immunisation such as family socioeconomic status, mothers literacy level, husbands permission before immunisation, religious beliefs about vaccines, rumours

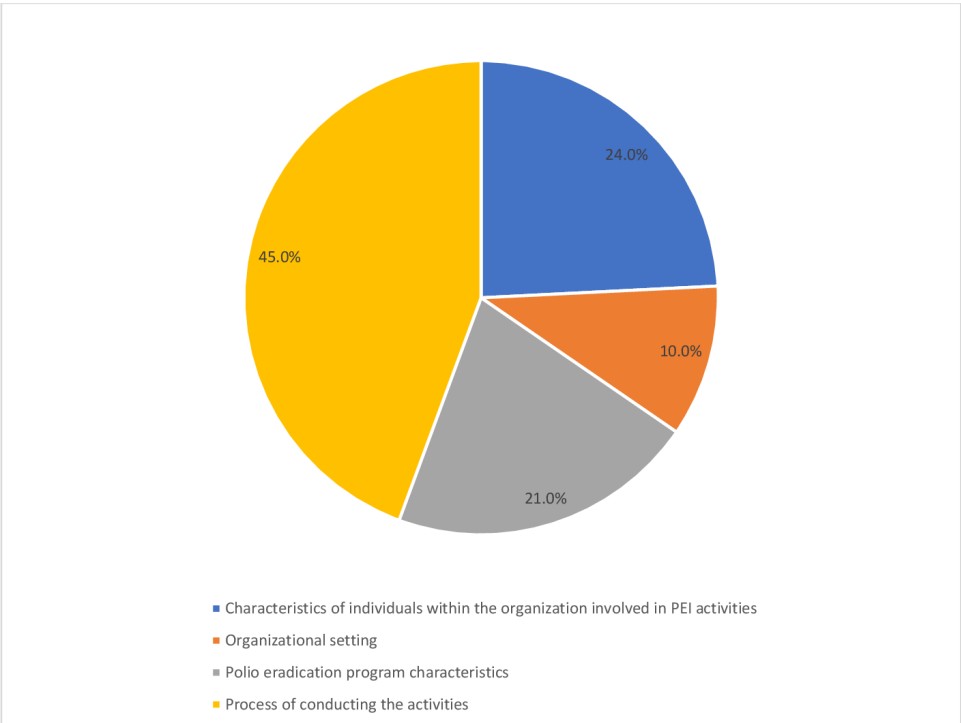

**Figure 1** Internal contributors to the successful implementation of community engagement activities.

and misconception surrounding vaccines, side effects of vaccination), served as a major barrier to community engagement. Hence, when properly harnessed, the social environment can contribute to the success of the community engagement goals but if neglected can hinder the successful implementation of a community

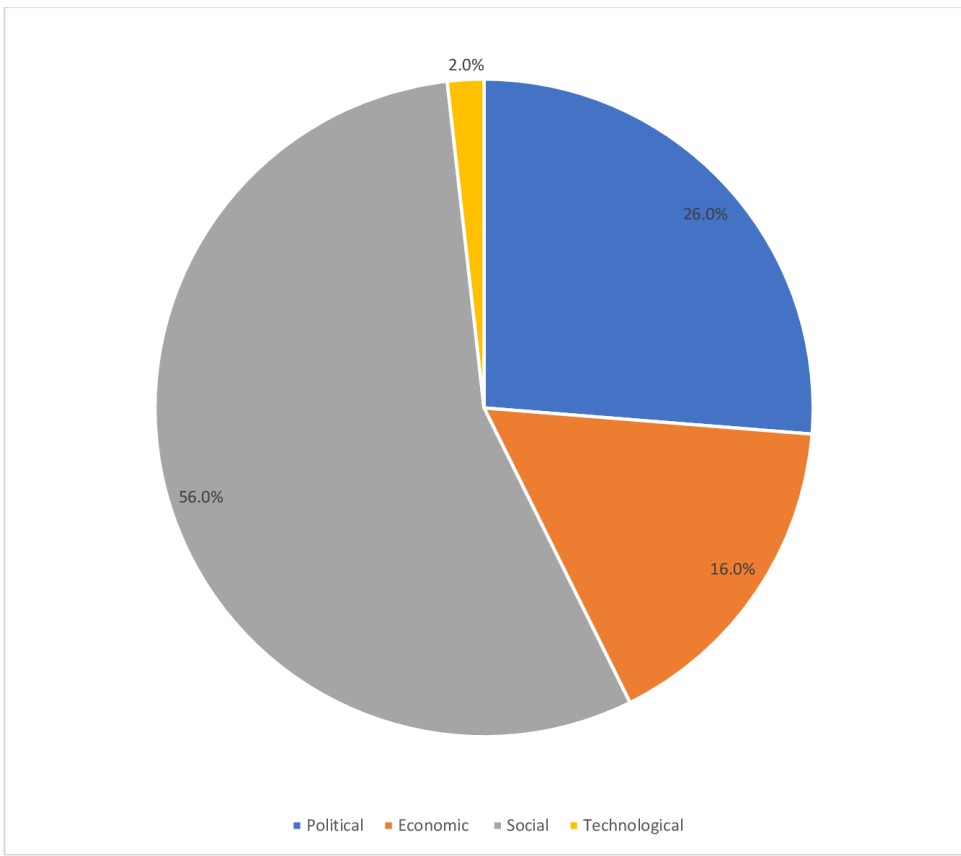

**Figure 2** External contributors to the successful implementation of community engagement activities.

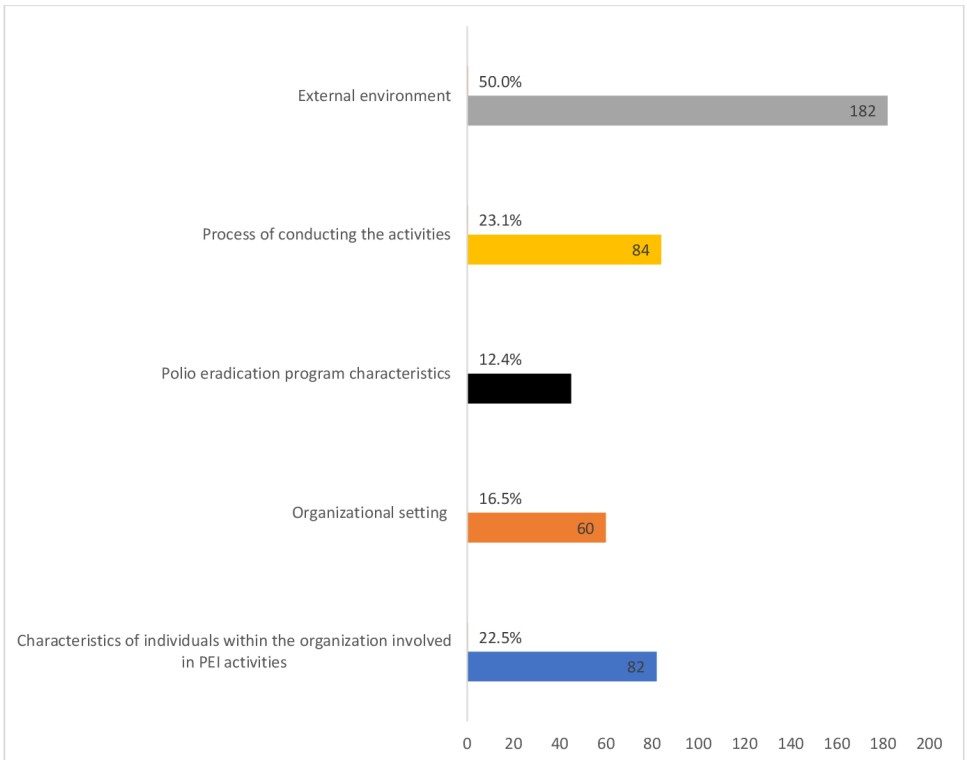

**Figure 3** Factors that impede the implementation of community engagement activities.

engagement strategy, just like many authors have reported.[28 29]

To harness these factors (process of conducting activities, individual characteristics, socio and economic environment), it is necessary to think about how readily modifiable these factors are. The process of conducting community engagement is a readily modifiable factor as it has evolved into a strategy that is led by the community members to ensure sustainability and ownership of the engagement process. While the characteristics of the PEI workforce have been modified over the years by equipping them with skills and abilities to provide sustained support to community engagement,[20] the sociocultural realities of a community are not readily modifiable. The PEI has successfully highlighted aspects of the culture that supports vaccination and addresses myths, beliefs and misconceptions that hinder participating in the immunisation exercises.[30] Therefore, finding a balance between the positives and negatives of any community's culture should be the goal of community engagement activities and not modifying the community's culture.

In the internal setting, there is an interplay of vital factors such as the process of conducting activities and characteristics of individuals within the organisation. According to Damschroder *et al*, essential components for the success of community engagement activities include the process of conducting activities that involved the planning of the activities, the mode of engagement and the point of entry into the community.[19] An instance of this interplay is an unmotivated health worker given a poorly planned programme to implement, this increases the possibility of failure in implementation failure. Other essential considerations for successful community engagement were: the characteristics of individuals to involve and how the community engagement is executed, monitored and evaluated.[19] The polio programme has documented the importance of measuring staff performance for the accountability framework and how it plays a role in achieving effective acute flaccid paralysis surveillance. Therefore, it is believed that harnessing staff performance which falls in the category of characteristics of individuals involved for community engagement will show similarly positive results.[31]

Based on the individuals involved in the programme, reflections on Africa's polio campaigns show, that any initiative is only as good as the quality of the human resource engaging with the community.[32] The PEI focused on local champions called the volunteer community mobilisers who advocate and align with the goal of the programme. They were purposeful in addressing community challenges (non-compliant households, vaccine-rejection cases, missed children) in Northern Nigeria.[33] Engagement of these individuals in the polio programmes, including religious and traditional leaders has aided the programme to address their misconceptions about the benefits of the polio programme and improving their self-efficacy to share information on polio immunisation has increased a sense of ownership of the polio programme which was the primary goal of community engagement, encouraged the participation of religious and traditional leaders in the programme and developed social networks within the community

**Table 3** Categories of factors that impede the implementation of community engagement activities*

| Factors | Categories | n (%) |
|---|---|---|
| External environment | Economic | 65 (35.7) |
| | Social | 60 (32.9) |
| | Political | 49 (26.9) |
| | Technological | 17 (9.3) |
| | Other environments | 53 (29.1) |
| Processes involved in conducting PEI programme activities | Engagement phase | 39 (46.4) |
| | Execution phase | 37 (44.0) |
| | Planning phase | 30 (35.7) |
| | Reflection phase | 23 (27.4) |
| Intervention/ PEI programme characteristics | Intervention source | 17 (37.8) |
| | Adaptability of the programme to the local context | 17 (37.8) |
| | Cost | 14 (31.1) |
| | Evidence strength and relative advantgae | 12 (26.7) |
| | Relative advantage | 12 (26.7) |
| | Complexity | 10 (22.2) |
| | Design quality | 9 (20.0) |
| | Trialability | 5 (11.1) |
| | Level of readiness for implementation | 28 (46.7) |
| Inner/organisational settings | Networks and communication | 21 (35.0) |
| | Culture | 17 (28.3) |
| | Implementation of climate | 16 (26.7) |
| | Structural charateristics | 11 (18.3) |
| Characteristics of individuals involved within the organisation | Individual's knowledge and beliefs about the activity | 51 (62.2) |
| | Individual's identification with the organisation | 23 (28.0) |
| | The individual's stage of change | 22 (26.8) |
| | Self-efficacy | 11 (13.4) |
| | Other personal attributes | 4 (4.9) |

*Multiple responses.
PEI, Polio Eradication Initiative.

will be available to work on routine immunisation and other health services as part of the polio legacy.[6]

The external setting, the social environment was identified as the most influential factor in supporting as well as undermining the community engagement process. In line with the submissions of several authors,[28 29] the unique context, culture, beliefs, modes of communication of any community could be a threat or facilitator of community participation in any programme. In the case of the polio programme in the northern part of Nigeria, the culture of preventing visitors contact with their wives, beliefs about the content of the polio vaccines, their nomadic way of life and their social norms about immunisation activities negatively affected the coverage rates in these areas.[13 34 35] However, engagement of the community members resulted in the identification of the problems and proffering of mutually acceptable solutions.[36]

This resistance to immunisation due to misconceptions and cultural beliefs in northern Nigeria was not as pronounced in the other parts of the country.[13 34] This shows a sharp contrast between the northern and the southern regions of the country.[13] In essence, this shows that for health interventions to be successful, there should be some measure of contextual basis to its implementation. The sociocultural context of communities should be considered and extensive dialogue carried out to ensure communities are adequately represented and take ownership of the programme. Similar to these, findings from previous studies on implementation research have also emphasised the importance of an intervention being contextual and flexible enough to be tailored to fit different environments.[27 37]

This study also highlighted another external factor, the political environment of the community. The PEI programme in Nigeria has over the years emphasised the importance of the political context in the success of any strategy both at the national scale and down to the grass-root level. For the PEI to succeed, it was important to win over political factions to increase the political will of the community representatives, avoid political interference in its activities so to make sure all community members benefit from the programme regardless of political factions.[30 38] Therefore, in addition to understanding the social context of the community, it is necessary that any community engagement seeks to understand the politics within the community, the various political factions with their beliefs and grievances. Most importantly, however, implementers must strive to remain politically neutral, to ensure that no section of the community is unrepresented due to their political affiliations.

The economic environment which involved the availability of sufficient revenues to fund immunisation activities and the primary healthcare system were the most influential external environmental factor considered by implementers of community engagement activities. For the polio programme, the Interagency Coordinating Committee and other GPEI partners responsible for providing considerable funds to the programme. These funds were used to set up 16 fully equipped laboratories to process stool and environmental samples both for the polio programme and other vaccine-preventable diseases and support the primary health system to deliver other maternal and child health interventions. The provision of health infrastructures is one of the legacies of the PEI.[39] Consequently, there have been adequate

funds for the successful implementation of the polio programme and the support community engagement in various communities for the success of their agenda. This factor shows that engaging community members goes beyond understanding the context of the community or just advocating for increased political will. It also involves funding to recruit community members as volunteers, to provide stipends or provide logistics support to religious and traditional leaders when they attend dialogue meeting or even providing incentives to caregivers. Therefore, future programme implementers should budget for the funding for community engagement activities and the sustainability of those funding for these activities even after the achievement of the project goals. This finding is corroborated by Andrus who highlighted the role of the funds in the success of programmes by stating that a constant demotivator for polio personnel across endemic countries has constantly been the lack of financial incentives.[33] While Bigna[24] summarised it in his words, 'Money is the nerve of war, so, going to war against wild poliovirus will require sufficient funding to destroy it'.

## CONCLUSION

The CFIR revealed specific contextual factors that influence the effectiveness of community engagement implementation. The process of conducting the activities and the external environment (economic, social and political) were very critical from this study. Programme implementers should collaborate transparently with the community, foster trust and maintain a relationship with the community. These findings contribute to implementation research literature supporting the use of the CFIR in displaying constructs and domains that influence health innovation implementation. These findings can be used by policymakers and researchers to advocate for effective ways to implement community engagement activities.

**Contributors** The research was conceived and designed by EO, OOAlonge, AK and OOAkinyemi, while data collection and analysis were implemented by EO, OOAkinyemi, AA, CN and CB. This manuscript was conceived by OOAkinyemi, AA and EO. The initial draft was developed by AA, CB and CN and reviewed by OOAkinyemi, AAF, AK, OA and EO. The final version of the manuscript was approved by OOAkinyemi, AA, CB, CN, AK, AAF, OOAlonge and EO.

**Funding** This study was funded by the Bill & Melinda Gates Foundation (Grant/Award Number: OPP1178578).

**Disclaimer** The funders did not have any role in study design, data collection and analysis, decision to publish, or preparation of the manuscript. The funder did not play any role in writing the protocol, interpreting the data, or in writing this manuscript. The funder provided and coordinated external peer-review for the study proposal. Publication costs were funded by the Bill and Melinda Gates Foundation.

**Competing interests** None declared.

**Patient and public involvement** Patients and/or the public were not involved in the design, or conduct, or reporting, or dissemination plans of this research.

**Patient consent for publication** Not required.

**Ethics approval** Ethical approval for this study was obtained from the National Health Research Ethics Committee (Approval Number: NHREC/01/01/2007-6/07/2018).

**Provenance and peer review** Not commissioned; externally peer reviewed.

**Data availability statement** Data are available on reasonable request. The data used and/or analysed in this study are available from the corresponding author on request.

**Open access** This is an open access article distributed in accordance with the Creative Commons Attribution 4.0 Unported (CC BY 4.0) license, which permits others to copy, redistribute, remix, transform and build upon this work for any purpose, provided the original work is properly cited, a link to the licence is given, and indication of whether changes were made. See: https://creativecommons.org/licenses/by/4.0/.

**ORCID iDs**
Oluwaseun Oladapo Akinyemi http://orcid.org/0000-0003-4135-1459
Anna Kalbarczyk http://orcid.org/0000-0002-6143-8634

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
