## [Reviewer comments · BMJ Open]

ARTICLE DETAILS

TITLE (PROVISIONAL)	Assessing community engagement in Nigeria polio eradication initiative: application of the Consolidated Framework for Implementation Research
AUTHORS	Akinyemi, Oluwaseun; Adebayo, Adedamola; Basse, Christopher; Nwaiwu, Chioma; Kalbarczyk, Anna; Fatiregun, Akinola; Alonge, Olakunle; Owoaje, Eme

VERSION 1 – REVIEW

REVIEWER	Oladoyin, Victoria
REVIEW RETURNED	27-Apr-2021

GENERAL COMMENTS	The research is satisfactory and will contribute to the body of knowledge. However, for this research to be repeatable, the methods needs further clarifications and reviews as pointed out in the body of the manuscript. The areas requiring correction is as highlighted in the body of the manuscript. Typographical and grammatical errors should be corrected. The reviewer provided a marked copy with additional comments. Please contact the publisher for full details.
---

VERSION 1 – AUTHOR RESPONSE

Manuscript ID bmjopen-2021-048694

Assessing community engagement in Nigeria polio eradication initiative: application of the Consolidated Framework for Implementation Research

RESPONSE TO REVIEWER'S COMMENTS

S/N	Reviewer's Comment	Response	Page
1	Of what? Please kindly clarify this.	Of poliomyelitis. Added	P 5, line 20
2	Please kindly recast.	Done as advised	P 5, line 22-28

3	It is not quite clear which of the recruitment process you are talking about here. Is it for the larger study or the subset of the larger study? Please kindly clarify this. I will also like to quickly state here that you should focus your write up on the subset of the larger study as this is the focus of this manuscript. The background information that this study was part of a larger study provided in the study design and participant section is enough. Please kindly focus your subsequent write-up, after the study design and participant section, on the current study.	This describes the recruitment process in the larger study. The section has been moved up and merged with the “Study design and participants” section. The header “Recruitment process” has been deleted.	P 6, line 1-7
4	Is it the larger study or this current study that utilized the CFIR as an evaluation framework to identify the critical challenges and contributors in the implementation of community engagement activities within the PEI in Nigeria? Please kindly clarify.	It is this current study. Thank you. This has been clarified	P 6, 9-11
5	This is not clear. Please kindly rephrase.	Done. The sentence has been rephrased.	P 6, line 12-15
6	Please kindly include the questionnaire you used for this study while responding to the reviewers’ comments. Please kindly provide detailed information on how the survey instrument was validated.	The questionnaire is attached. Face validation of the instrument was done through pretesting	P 7, line 5-17
7	From the larger study, those who met your inclusion criteria for this study were 364. It is not quite clear why you were aiming to reach a target of 1000 as this manuscript was borne out of a	The unnecessary information about the larger study has been deleted.	P 10, line 3-8
	sub-analysis of a larger study. You already knew those who met your inclusion criteria from the analysis of the larger study, so why were you aiming for 1,000?		
8	Does this imply that all your respondents have email addresses?	No. Some were reached via email while some others had the questionnaire administered face-toface. Please see the paragraph below for the explanation.	P 10, line 5
9	Recast	Done as advised	P 10, line 13-14

10	How did you handle duplicate data?	Duplicate data were deleted	P 10, line 17
	Please kindly include the NHREC number	NHREC number added	P 10, line 31
11	Do you mean electronic signatures? Please kindly clarify.	For those who completed the form online, Consent was provided electronically by entering their names in lieu of signatures.	P 11, line 1
12	Kindly use consistent terminologies in the result pros and the tables.	Everywhere subdistrict appeared, local government has been added for consistency	P 11, line 11
13	Please kindly clarify which level is defined as sub district in the methods section	Subdistrict refers to “Local Government Area”; this has been clarified	P 11, line 15, Table 2
14	Please kindly analyse the factors that promote the implementation of community engagement strategy at the level of the five domains combined as well as at the level of the questions that make up the domains, just like you did for the barriers in table 3 and figure 3.	Please note that the factors that facilitated he implementation of the community engagement strategy has been presented in Tables 1 to 2. Presenting the same data as a Table may not be desirable.	P 12, line 4-23
15	Please be consistent in your write-up. The facilitators were recategorized as internal and external factors, I think for the first time here. I will recommend you analyse and present your result at the level of the domains and questions that make up the domain as stated above. The concept of internal or external factor can come up in your discussion, if you so wish.	Please refer to Table 1 where the domains of the CFIR was presented, domains I – IV represents the “Internal settings” or “internal contributor”, whereas, domain V (with parts A to E) was the “External or Outer setting”. So, the classification is not just coming in the result. CFIR, as a framework, reverberates throughout the manuscript.	P 12, line 9
16	The result section is not a repetition of the tables and figures.	This section did not attempt to repeat the findings presented in the tables and figures but rather to help the readers make sense of them.	P 12, line 25-32
17	Please kindly discuss in line with your study objectives and results.	Kindly note that the discussion in line with the objectives and results.	P 15, Discussion section
		However, sometimes in order to promote clarity, the facilitator and barriers of each domain are discussed together.	
18	This statement is too long. Please kindly split it.	Statement has been split into two.	P 15, line 6-13
19	Recast	Done	P 15, line 22-25